# FGF-23 as a Biomarker for Carotid Plaque Vulnerability: A Systematic Review

**DOI:** 10.3390/medsci13010027

**Published:** 2025-03-10

**Authors:** Joana Oliveira-Sousa, Mariana Fragão-Marques, Luís Duarte-Gamas, Hugo Ribeiro, João Rocha-Neves

**Affiliations:** 1RISE-Health, Department of Biomedicine, Unit of Anatomy, Faculty of Medicine, University of Porto, 4200-319 Porto, Portugal; marianaif.rm@gmail.com (M.F.-M.); hribeiroff@gmail.com (H.R.); joaorochaneves@hotmail.com (J.R.-N.); 2Department of Angiology and Vascular Surgery, Local Health Unit Tâmega e Sousa, 4560-136 Penafiel, Portugal; afonso_gamas@hotmail.com; 3Faculty of Medicine, University of Coimbra, 3004-528 Coimbra, Portugal; 4Community Palliative Care Team Gaia–Local Health Unit Gaia and Espinho, 4434-502 Vila Nova de Gaia, Portugal; 5Coimbra Institute for Clinical and Biomedical Research, 3000-548 Coimbra, Portugal

**Keywords:** klotho, atherosclerosis, carotid endarterectomy, stroke risk, inflammation, plaque instability

## Abstract

Background/Objectives: Carotid artery disease is a condition affecting 3% of the general population which significantly contributes to the development of cerebrovascular events. Fibroblast Growth Factor-23 (FGF-23) is a hormone that has been linked to atherosclerosis and increased cardiovascular risk, including stroke and myocardial infarction. This review explores the association of FGF-23 with carotid artery disease progression in an endarterectomy clinical context. Methods: Based on Preferred Reporting Items for Systematic Reviews and Meta-Analyses (PRISMA), a search was performed relying on MEDLINE, Scopus and Web of Science, identifying publications focused on the correlation between serum FGF-23 and carotid artery disease. Assessment of study quality was made using National Heart, Lung and Blood Institute Study Quality Assessment Tool (NHLBI). Results: Three observational studies, comprising 1039 participants, were included. There was considerable heterogeneity among the populations from the different studies. Elevated FGF-23 levels were consistently associated with unstable plaque features, including intraplaque neovascularization, as identified through Superb Microvascular Imaging (SMI). Plasma levels of inflammatory mediators, such as Interleukin-6 (Il-6), Monocyte Chemoattractant Protein-1 (MCP-1), and Osteoprotegerin (OPG), positively correlated with carotid artery disease, but their link to unstable plaques is conflicting. None of the studies investigated clinical complications following carotid endarterectomy. Conclusions: FGF-23 is a potential biomarker for plaque vulnerability in carotid disease. Despite promising findings, limitations such as small sample sizes and lack of longitudinal data suggest the need for larger and more diverse studies to improve risk stratification and inform personalized treatment strategies for carotid atherosclerosis.

## 1. Introduction

Cerebrovascular events are among the leading causes of mortality worldwide, with carotid artery disease playing a critical role in increasing this risk [1]. In fact, 20–30% of all ischemic strokes are caused by ipsilateral carotid atherosclerosis [2]. Furthermore, more than half of stroke survivors experience long-term dependency, creating substantial burdens for healthcare systems and caregivers [3].

The atherosclerotic plaques formed within the carotid can evolve towards instability, increasing their susceptibility to rupture and thrombosis [4]. To mitigate this risk, carotid endarterectomy, a surgical procedure for plaque removal, was shown to be an effective intervention, particularly in patients with moderate and severe carotid stenosis [5].

Despite its efficacy, the peri- and post-operative complications of carotid endarterectomy depend on the characteristics of the plaques, particularly their vulnerability [6,7]. Therefore, understanding the factors that drive the progression of atherosclerotic plaques towards instability is crucial for improving clinical outcomes in patients undergoing this procedure. One of the contributing factors to plaque vulnerability is the development of neovascularization. Other risk factors include cap thinning, the expansion of the lipid core, the presence of microcalcifications, proteases that aid in matrix digestion and elevated levels of inflammatory cytokines [8].

Fibroblast growth factor-23 (FGF-23), a 251-amino-acid protein synthetized and secreted by osteoblasts and osteocytes, is a phosphotropic hormone, contributing to phosphate homeostasis [9], and has also been implicated in the progression of carotid artery disease [10,11,12]. FGF-23 exerts its effects through its obligate co-receptor, Klotho, which is expressed in human vascular tissues [13,14]. Together with FGF-receptors (FGFR1c, FGFR3c, FGFR4), a trimeric signaling complex is formed, mediating the actions of FGF-23 [15].

Despite its physiological functions, it has been demonstrated that FGF-23 is involved in systemic atherosclerosis via nitric-oxide-associated endothelial dysfunction and calcium-phosphate-related bone mineralization [16].

Initially, studies were performed on patients with Chronic Kidney Disease (CKD)—as CKD progresses, phosphate excretion declines, leading to a compensatory rise in FGF-23 serum levels to maintain phosphate balance [17]. As Klotho expression is decreased in CKD, persistent increase in FGF-23 levels was shown to contribute to endothelial damage and cardiovascular events [18].

Furthermore, FGF-23 levels in the serum were also associated with a higher risk of cardiovascular disease (CVD), regarding ischemic stroke, myocardial infarction and heart failure, and these associations were not limited to patients with impaired kidney function [19,20]; FGF-23 levels were shown to have a more significant burden of carotid atherosclerosis independent of CKD [21].

Building on its established role in systemic atherosclerosis, recent studies have explored the potential involvement of FGF-23 in carotid disease pathophysiology [11,12,22,23,24,25].

This systematic review aims to evaluate the relationship between elevated FGF-23 levels and the characteristics of the atherosclerotic plaque in patients undergoing endarterectomy. Secondary objectives include exploring correlations between inflammatory mediators and FGF-23 serum levels, as well as exploring its relationship with clinical complications after carotid endarterectomy.

## 2. Materials and Methods

This systematic review was carried out following the Preferred Reporting Items for a Systematic Review and Meta-analysis (PRISMA) Statement [26] and the protocol has been registered in Prospero (reference: CRD42024605818).

Observational studies involving patients subjected to carotid endarterectomy (CEA) were included. To be eligible, the studies were required to report FGF-23 levels before and/or after surgery and evaluate carotid plaque characteristics or any perioperative outcomes. Studies were excluded if they were animal studies, case reports, reviews, commentaries, editorials, conference abstracts, or non-peer-reviewed publications. The publication language and date were not considered as criteria for exclusion in this review.

To perform this review, a search in three electronic databases—MEDLINE, Scopus and Web of Science—was performed by J.O.S and J.R.N in October 2024. Additionally, reference lists of the included studies were screened by J.O.S to identify further potentially relevant articles.

A search query was used to identify potential articles to be included in this review and is presented in Appendix A. In MEDLINE, the query used was as follows: (“Endarterectomy” [MeSH Terms] OR “Endarterectomy, Carotid” [MeSH Terms] OR “Carotid Endarterectomies” [MeSH Terms] or “Carotid Endarterectomy” [All Fields]) AND (“Fibroblast Growth Factor-23” [MeSH Terms] OR “FGF23” [All Fields] OR “Fibroblast Growth Factor 23 [All Fields]”), with adaptations for further use in the other databases.

Initially, duplicate studies were eliminated by J.O.S. Subsequently, titles and abstracts of the preselected studies were screened in an independent manner by two reviewers (J.O.S and J.R.N) and, after exclusion, the remaining articles were analyzed by the same reviewers in full text to determine the accomplishment of the previously established eligibility criteria.

Study identification, including publication year, journal of publication, study design, study center and sample size, along with the population demographics, namely the participants’ type, age, gender distribution and number of patients submitted to CEA, as well as the frequency of cardiovascular comorbidities were analyzed and retrieved by two reviewers (J.O.S and J.R.N). Additionally, data related to procedural protocols was retrieved, as well as the criteria used to define complicated plaques and information regarding FGF-23 and the inflammatory mediators serum levels, with the accompanying statistical analysis.

Serum levels of FGF-23 and inflammatory markers were presented as mean [23] and median [22,23,24], accompanied by the respective confidence interval [23] or interquartile range [22,24]. Correlation analyses between continuous variables, including FGF-23 and inflammatory mediators levels, were assessed using χ2 test for categorical values [22], Kruskal–Wallis one way ANOVA followed by a Dunn posttest [23], post hoc pairwise comparison and Spearman ρ correlation [24].

The study quality was obtained independently by J.O.S and J.R.N, using the National Heart, Lung and Blood Institute (NHLBI) Study Quality Assessment Tool for observational cohort and cross-sectional studies (2021) [27]. The quality of evidence for the included articles was evaluated using the Grading of Recommendations, Assessment, Development, and Evaluation (GRADE) approach. Articles were classified into four levels of quality (high, moderate, low, and very low) [28].

During this process, any discrepancies between J.O.S and J.R.N were solved by a third reviewer, namely M.F.M.

## 3. Results

### 3.1. Search Results

The search retrieved 20 articles, of which 11 were screened by title and abstract after elimination of duplicates. Through screening, six were considered eligible for full text appraisal and three were included in this review, comprising a total of 1039 patients [22,23,24]. Through a first screening of abstracts of potential includable studies, the reasons for exclusion were the following: irrelevant topic (n = 3) and wrong population (n = 2). The definitive reasons for exclusion, identified through comprehensive full-text assessment, were as follows: basic research study (n = 1) and inexistence of carotid endarterectomy (n = 2) (Figure 1).

### 3.2. Description of Studies

All of the articles included in this review were observational cohorts, one of them being retrospective [22] and the other two cross-sectional [23,24]. The characteristics of each study and its population are displayed in Table 1. The studies were performed in two different countries in Europe, namely Italy [22,23] and Norway [24]. The sample sizes varied among the studies reviewed. Biscetti et al. included 959 participants in their study [22], whereas Del Porto et al. and Zamani et al. had sample sizes of 51 and 29 individuals, respectively [23,24]. The mean participants’ age also differed across studies. Biscetti et al. reported 72.05 years [22], Del Porto et al. documented 71.46 years [23] and Zamani et al. recorded 72.5 years [24]. The percentage of male patients across studies was 36.8% in one study [22], and 62.1% [24] and 80.71% [23] in the other two studies. The prevalence of symptomatic carotid stenosis was reported as 18.1% by Biscetti et al. [22], 17.6% in Del Porto et al.’s study [23] and 65.5% in Zamani et al.’s study [24]. The comorbidities of the populations included in the studies were collected and presented in Table 2.

The criteria employed to characterize atherosclerotic plaques varied across studies. Zamani et al. [24] used a modified version of the Gray–Weale Classification for ultrasound imaging [29], complemented by the assessment of Intraplaque Microvascular Flow (IMVF) using Superb Microvascular Imaging (SMI). The IMVF was categorized into three distinct levels based on a visual scale to define qualitatively intraplaque neovascularization (IPN). Furthermore, a visual evaluation of IMVF signals was conducted, quantifying the number of neovessels identified within a 2-min SMI videoclip and therefore providing a quantitative measure of IPN [24]. The other two studies used the American Heart Association (AHA) criteria to define complicated and uncomplicated plaques [22,23]. Furthermore, the stenosis cutoff used also varied between studies. Two articles considered >70% to be the cutoff to include patients with internal carotid stenosis [22,23]. Conversely, another study included patients with >50% carotid stenosis [24].

### 3.3. Main Findings

#### 3.3.1. Primary Outcomes

Serum levels of FGF-23 and inflammatory mediators, along with the corresponding statistical analyses comparing subgroups in each study, are presented in Table 3 and Table 4, respectively. Serum levels of FGF-23 were significantly higher in patients with carotid artery disease and, more specifically, unstable plaques, in two of the selected studies [22,23]. Similarly, in the remaining included study, FGF-23 circulating levels were significantly higher in the presence of plaques with higher grades of IMVF signal on the semi-quantitative-SMI visual scale (*p* = 0.011, r = 0.466)—patients in the extensive-IPN group had increased plasma levels of FGF-23 compared to the other patient groups—and higher neovessel count recorded by quantitative SMI (*p* = 0.007, r = 0.491) [24]. Del Porto et al. further demonstrated a significant increase in FGF-23 expression in complicated plaques compared to uncomplicated plaques (*p* < 0.025). Monocytes/macrophages (CD68+) were present in 90% of the complicated plaques, whereas only 53.3% of the uncomplicated plaques exhibited these cells. Additionally, a positive cytoplasmic reaction for FGF-23 was observed in both fibroblasts and monocytes/macrophages within the plaques, which was confirmed through double-staining immunofluorescence, particularly in CD68+ cells [23]. The covariates used in the multiple regression models are presented in Appendix A. FGF-23 was consistently and independently associated with unstable plaques, in patients with internal carotid artery stenosis (ICAS), and SMI-assessed intraplaque neovessel count.

#### 3.3.2. Secondary Outcomes

The serum levels of inflammatory mediators, namely High Sensitivity C Reactive Protein (HsCRP), OPG [22], IL-6 [22,23], MCP-1 and Vascular Endothelial Growth Factor (VEGF) [23] were also measured. Biscetti et al. demonstrated that circulating levels of HsCRP, OPG and IL-6 were significantly higher in patients with internal carotid artery stenosis (ICAS) and, particularly, unstable plaques [22]. Del Porto et al. indicated a positive correlation between IL-6, Interleukin-8 (IL-8) and MCP-1 serum levels and critical carotid artery stenosis, although no significant correlation was found between the serum levels of these mediators and complicated plaques. Besides this, VEGF correlations were controversial: VEGF levels were significantly higher in patients with uncomplicated plaques when compared with controls, but not significantly different between patients with complicated plaques and either controls or those with uncomplicated plaques [23]. On the other hand, Zamani et al. demonstrated no correlation between the levels of basic fibroblast growth factor (b-FGF), Il-6, Tumor Necrosis Factor α (TNF-α) and C Reactive Protein (CRP) and the semi-quantitative and quantitative SMI-assessed IPN. Moreover, no correlation was found between FGF-23 and CRP serum levels (*p* = 0.999, r = 0) [24].

No article underwent an appraisal of short and long-term complications of carotid endarterectomy and compared it to FGF-23 plasma levels.

### 3.4. Quality of Studies

The risk of bias of the selected articles is displayed in Figure 2 and Figure 3. The risk of bias for each individual study is presented in Figure 2, while the overall assessment for each evaluated item across the studies is illustrated in Figure 3. All the studies included in this review had an overall average/high risk of bias. The main concerns were patient recruitment, sample size justification, exposure assessment and confounding variables adjustment.

## 4. Discussion

This systematic review synthesizes findings from three studies, examining FGF-23 in carotid artery disease in patients submitted to endarterectomy.

The key finding of this review was the positive association between higher FGF-23 serum levels and unstable carotid plaque characteristics, including neovascularization, as identified through SMI, hindering FGF-23 as a potential biomarker for plaque vulnerability. Additionally, while inflammatory mediators such as IL-6, MCP-1, and OPG were positively correlated with carotid artery disease, their relationship with plaque instability was inconsistent among studies.

Fibroblast Growth Factor-23 (FGF-23) has emerged as a biomarker in cardiovascular research [30,31]—previous studies have shown that FGF-23 is an independent predictor of cardiovascular events in the community [32,33,34] and its association with atherosclerosis has been established [35,36], though its exact role remains unclear.

In coronary artery disease, several trials have highlighted the relevance of FGF-23. Indeed, high levels of serum FGF-23 were associated with increased risk of coronary artery disease, in a cohort study involving 11,638 participants [37]. Supporting these findings, a positive correlation between FGF-23 serum levels and coronary calcifications [35,38], as well as the extent of coronary artery stenosis [39], has been previously identified.

Consistently across the studies included in this review, elevated FGF-23 levels were significantly associated with unstable carotid plaques [22,23], underscoring its potential as a biomarker for atherosclerotic vulnerability. These findings align with prior evidence from population-based studies, which have shown that high levels of FGF-23 are implied in presence and burden of cerebral atherosclerosis in stroke patients [12], as well as in plaque presence, size and increased vessel wall intima-media thickness in stroke-free patients [20,21]. Regarding the underlying pathophysiological mechanisms, FGF-23 can disrupt plaque stability through various mechanisms. On one hand, in experimental models, FGF-23 reduces nitric oxide bioavailability in the endothelium by promoting the synthesis of superoxide anions [40], impairing endothelial-dependent vasodilatation [41] and leading to endothelial damage.

On the other hand, FGF-23 is capable of modulating the release of inflammatory mediators [42,43,44,45,46,47,48] through klotho-independent effects [42], which in turn contribute to atherosclerotic disease progression [49]—vulnerable plaques are characterized by an “active” inflammation [50]. However, in this review, even though in two of the included studies [22,23] inflammatory mediators were found to have a positive correlation with carotid disease, the correlation between inflammatory markers and unstable plaques was contradictory. Biscetti et al. found a significant relationship between circulating levels of IL-6, OPG and HsCRP and unstable plaques [22]. On the other hand, Del Porto et al., despite demonstrating a greater degree of infiltration by monocytes and macrophages in atherosclerotic lesions, found no correlation between serum levels of IL-6, IL-8, MCP-1 and VEGF and complicated plaques. Furthermore, Del Porto et al. identified the presence of FGF-23 within CD68+ macrophages in critical carotid atherosclerotic lesions [23], emphasizing the positive relationship between inflammation and FGF-23; in fact, inflammation is associated with increased FGF-23 in many different diseases [42,51,52,53,54]. More specifically, TNF-α, Interleukin 1B (IL-1B), and IL-6 have been shown to dose-dependently upregulate FGF-23 gene expression [55,56,57,58,59,60,61]. Collectively, these findings support a possible positive bidirectional relationship between FGF-23 and serum levels of inflammatory mediators, with possible therapeutic implications.

Another mechanism through which FGF-23 may disrupt plaque stability is neovascularization. Hypoxia is one of the major drivers of neovascularization [62,63], by inducing the hypoxia-induced transcription factor 1-α (HIF1-α) signaling pathway [64]. In humans, hypoxia and HIF1-α were previously shown to lead to an increase of FGF-23 levels [65,66,67]. In one of the studies included in this review [24], in opposition with b-FGF, a positive association between FGF-23 serum levels and SMI-assessed IPN was found, implying its possible role in the process of plaque neovascularization development. In contrast, Zamani et al. identified no relationship between TNF-α, IL-6 and CRP with SMI-assessed intra-plaque neovascularization (IPN), and no correlation between CRP and FGF23 serum levels [24]. However, inflammation has been highlighted as a potent trigger of neovascularization in different settings [68,69]. Regarding CRP, some authors have observed that it inhibits VEGF production and angiogenesis [70,71], even though it was shown that CRP upregulates VEGF expression [72], contributing to neovessel formation.

The association between plaque instability and perioperative stroke and death has been reported in a previous study [73]. Furthermore, plaque vulnerability was also associated with an increased number of embolisms during the perioperative period after CEA [74,75]. Current guidelines for carotid endarterectomy emphasize stenosis severity [76,77,78]. However, with the growing evidence that the risk of stroke is not equivalent for all patients with similar grades of stenosis, the aforementioned guidelines are continuously being challenged [79,80]—emerging evidence suggests that plaque morphology should be a key factor in selecting patients for revascularization. In fact, the European Society of Vascular Surgery (ESVS) has identified clinical and imaging features linked to an increased risk of late stroke in patients with asymptomatic carotid stenosis (ACS) of 50% to 99%. Also, data from the Athero-Express study revealed a higher risk of late cardiovascular events in patients with abundant neovascularization at CEA [81]. Therefore, based on our findings that elevated FGF-23 levels are associated with unstable plaques and neovascularization, we can hypothesize that elevated FGF-23 levels may also be associated with a higher incidence of post-endarterectomy complications. However, future clinical trials are necessary to validate this hypothesis, since it is a promising cheap and easy to obtain biomarker.

This review is subject to several potential limitations. The studies reviewed were observational, limiting causal inference. Additionally, the small sample sizes in the studies included, retrieved from predominantly elderly, Caucasian populations reduce statistical power and the generalizability of the findings. On the other hand, heterogeneity between study populations further hinders the interpretation of results: while Biscetti et al. focused exclusively on type 2 diabetic patients, Zamani et al. and Del Porto et al. included broader populations. However, the single-center nature of all the included studies introduces potential selection bias, further limiting the external validity of findings. Measurement variability and methodological challenges also warrant consideration. First, the criteria for defining complicated or unstable plaques varied, limiting comparability across studies. Secondly, the time points for FGF-23 measurement also varied: while in two studies, FGF-23 was measured at a single pre-operative time point [22,24], Del Porto et al. [23] measured it at multiple time points.

The definition of plaque instability varied across studies. Biscetti et al. [22,24] and Del Porto et al. [22,24] classified plaques as unstable based on histopathological criteria from the American Heart Association (AHA), whereas Zamani et al. [24] relied on intraplaque neovascularization (IPN) assessed via Superb Microvascular Imaging (SMI). This divergence suggests that some plaques exhibiting high inflammatory activity may not necessarily display increased neovascularization, leading to discordant findings. Biscetti et al. exclusively studied diabetic patients, who typically exhibit heightened systemic inflammation, while Zamani et al. examined a broader, more heterogeneous population. This difference in patient selection might contribute to variations in the association between inflammatory markers and plaque instability. Variability in the measurement of inflammatory biomarkers and FGF-23 levels could also play a role.

The potential interplay between inflammation and hypoxia in driving plaque instability remains a critical avenue for future investigation. Given the growing evidence linking FGF-23 to neovascularization, further research should explore whether hypoxia-driven pathways are independent or synergistic with inflammatory processes.

Besides this, the covariates used in adjusted models also varied among studies, which raises the possibility of the presence of different confounders. Potential confounding factors like diabetes mellitus, CKD, vitamin D and calcium metabolism or medications varied among the study populations, potentially decreasing the ability to form independent conclusions in predicting the association of FGF-23 with plaque instability. Furthermore, the potential for publication bias cannot be ignored, given the limited number of studies and the likelihood that positive findings were preferentially published. Finally, a significant drawback of the studies included in this review is the absence of long-term follow-up. Without monitoring of FGF-23 levels after carotid endarterectomy, it remains uncertain if this biomarker can predict restenosis, recurrent stroke, or long-term cardiovascular mortality. Upcoming research should include follow-up durations of no less than 6–12 months to evaluate the risk of restenosis and 1–5 years to examine the recurrence of strokes and survival results.

Despite the limitations, serious clinical implications are suggested by this review. On one hand, FGF-23 might reveal itself as a biomarker for identifying patients with carotid disease (at higher risk of adverse outcomes), and its integration into clinical practice could enhance decision-making regarding the timing and appropriateness of carotid endarterectomy (CEA), particularly in asymptomatic patients. Additionally, monitoring FGF-23 levels in the postoperative period could provide additional insights into the risk of complications, such as restenosis or recurrent stroke, allowing for tailored postoperative care. On the other hand, anti-inflammatory treatments or therapies targeting neovascularization may serve as adjunctive measures to improve outcomes in patients undergoing CEA. To test these hypotheses, study design involving prospective, multicenter, 1–5 year follow-up studies, with patients stratified by CKD, diabetes mellitus and statin use, having primary endpoints such as restenosis, stroke and cardiovascular mortality, and secondary endpoints such as endothelial dysfunction, inflammation, and intraplaque neovascularization, using multivariate adjustment and propensity score matching in statistical analysis should be encouraged, as it could help in gathering more information about clinical implications of FGF-23.

## 5. Conclusions

This systematic review identifies FGF-23 as a potential biomarker for plaque vulnerability, including plaque neovascularization in patients with carotid disease undergoing carotid endarterectomy. Elevated FGF-23 levels correlate with unstable plaque features, including neovascularization and inflammation, linking it to atherosclerotic progression. However, as of now it is not possible to establish any level of recommendation regarding FGF-23’s clinical application. Further research should aim at larger and broader populations, in a prospective manner, further exploring these correlations, as well as short and long-term complications of carotid endarterectomy. These insights could enhance risk stratification and personalized treatment strategies for carotid atherosclerosis.

## Figures and Tables

**Figure 1 medsci-13-00027-f001:**
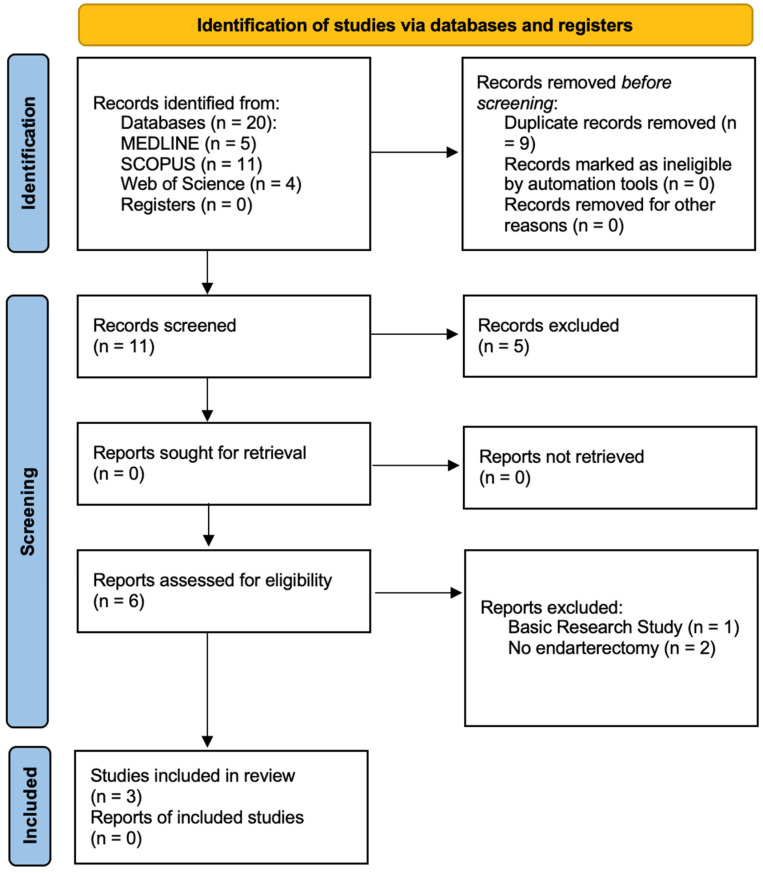
PRISMA flow chart regarding the process of identification and selection of the studies.

**Figure 2 medsci-13-00027-f002:**
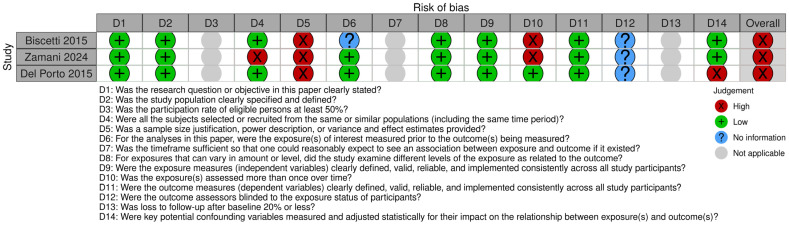
Risk of bias of the observational studies included in this systematic review, displayed by article.

**Figure 3 medsci-13-00027-f003:**
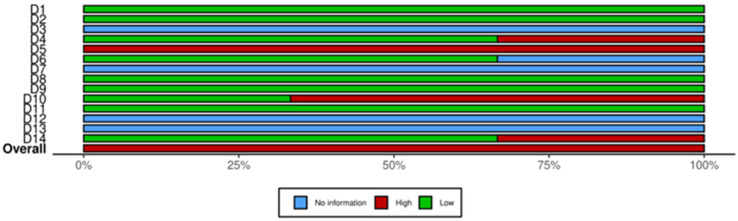
Risk of bias of the observational studies included in this systematic review, displayed by item.

**Table 1 medsci-13-00027-t001:** Identification and overview of each study.

Study ID						
Study	Publication Year	Journal	Study Center	Study Design	Sample	GRADE
Biscetti, F. et al. [22]	2015	*Cardiovascular Diabetology*	A. Gemelli University Hospital, Catholic University School of Medicine, Rome, Italy	Retrospective Observational Study	959	**
Del Porto, F. et al. [23]	2015	*Internal and Emergency Medicine–Official Journal of the Italian Society of Internal Medicine*	Sant’ Andrea Hospital, “Sapienza” University of Rome, Italy	Cross-Sectional Observational Study	51	*
Zamani, M. et al. [24]	2024	*Frontiers in Immunology*	Oslo University Hospital, Oslo, Norway	Cross-Sectional Observational Study	29	*
**Population**						
**Study**	**Type of Participant**	**Recruitment Time**	**Age, Mean (Years)**	**Male, %**	**Number of CEA**	**Follow-Up**
Biscetti, F. et al. [22]	DM2	January 2009–February 2015	72.05	36.8	361	-
Del Porto, F. et al. [23]	-	NA	71.46	80.71	35	-
Zamani, M. et al. [24]	-	Up to January 2019/January 2019–April 2019	72.5	62.1	NA	-

Legend: CEA—Carotid Endarterectomy; DM2—Diabetes Mellitus Type 2; NA—Non-Available Information; */**—GRADE Summary Evaluation of Evidence Quality (1–4).

**Table 2 medsci-13-00027-t002:** Comorbidities of the population samples for each study.

Study	Hypertension, n (%)	Dyslipidemia, n (%)	Diabetes Mellitus, n (%)	Smoking History, n (%)	Coronary Artery Disease, n (%)	Carotid Territory Symptoms/Signs, n (%)
Biscetti, F. et al. [22]	482 (50.3)	460 (48.0)	959 (100.0)	761 (79.4)	392 (40.9)	174 (18.1)
Del Porto, F. et al. [23]	35 (68.6)	16 (31.4)	9 (17.6)	9 (17.6)	NA	9 (17.6)
Zamani, M. et al. [24]	18 (62.1)	13 (44.8)	4 (14.8)	13 (44.8)	11 (37.9)	19 (65.5)

Legend: NA—Non-Available Information.

**Table 3 medsci-13-00027-t003:** Serum levels of FGF-23 and inflammatory mediators in different groups of each study.

Study	Comparison Groups (Patients)	Serum Levels (Mean/Median ± SD/IQR)							
		FGF23 (pg/mL)		HsCRP (mg/L)	OPG (pmol/L)	IL-6 (pg/mL)	IL-8 (pg/mL)	MCP-1 (pg/mL)	VEGF (pg/mL)
Biscetti, F. et al. ^‡^ [22]	ICAS (n = 361)	67.7 [59.5–77.8]		7.95 [6.48–9.55]	4.84 [3.52–5.95]	62.3 [57.1–68.5]	NA	NA	NA
	Controls (n = 598)	43.89 [37.5–50.4]		3.95 [2.34–5.31]	2.48 [1.69–3.35]	39.1 [33.8–45.2]	NA	NA	NA
	Unstable Plaque (n = 166)	78.4 [68.4–87.5]		9.31 [7.94–11.1]	6.04 [4.65–7.34]	71.5 [66.3–77.4]	NA	NA	NA
	Stable Plaque (n = 195)	34.7 [29.7–41.1]		2.74 [1.92–4.12]	2.12 [1.02–2.95]	32.6 [28.8–36.6]	NA	NA	NA
Del Porto, F. et al. ^£^ [23]	A ^†^ (n = 20)	t0 *:	7.20 (7.45 ± 3.05) ^Δ^	NA	NA	114 ± 60.19	49.3 ± 9.71	391.9 ± 126.39	176.45 ± 239.28
		t1 **:	11.18 (12.13 ± 5.80) ^Δ^	NA	NA	NA	NA	NA	NA
	B ^†^ (n = 15)	t0 *:	1.80 (2.61 ± 1.76) ^Δ^	NA	NA	104.33 ± 50.59	46.56 ± 5.12	363.14 ± 121.17	270.7 ± 155.20
		t1 **:	3.35 (3.90 ± 2.52) ^Δ^	NA	NA	NA	NA	NA	NA
	C ^†^ (n = 16)	1 (2.53 ± 3.3)		NA	NA	3.99 ± 1.25			
Zamani, M. et al. ^¶^ [24]	No IPN (n = 6)	84.2 [55.8–136.2]		NA	NA	NA	NA	NA	NA
	Moderate IPN (n = 9)	78.1 [58.5–112.7]		NA	NA	NA	NA	NA	NA
	Extensive IPN (n = 14)	132.8 [64.9–236.5]		NA	NA	NA	NA	NA	NA

* t0: just prior to carotid endarterectomy; ** t1: 30 min after carotid endarterectomy; ^Δ^ FGF-23 serum levels were only measured in 12 patients with complicated plaques and seven patients with uncomplicated plaques; ^†^ A: Complicated Plaques; B: Uncomplicated Plaques; C: Controls; ^‡^ FGF3 serum levels were measured using a second-generation C-terminal human enzyme-linked immunosorbent assay (Immutopics, San Clemente, CA, USA); ^£^ FGF3 serum levels were measured by Enzyme-Linked Immunosorbent Assay (Immunotopics, Inc); ^¶^ FGF3 serum levels were analyzed by U-Plex Metabolic Group 1 (human) assay (Meso Scale Diagnostics, Rockville, MD, USA); Legend: ICAS—Internal Carotid Artery Stenosis; IPN: Intraplaque neovascularization; NA—Not Available.

**Table 4 medsci-13-00027-t004:** Statistical analysis of the different comparison groups.

Study	Comparison Groups (Patients)	Statistical Analysis (Serum Levels)										
		FGF23		HsCRP	OPG	IL-6	IL-8	MCP-1	VEGF	TNF-α	b-FGF	CRP
Biscetti, F. et al. [22]	ICAS (n = 361)	*p* < 0.001 ^‡^		*p* < 0.001 ^‡^	*p* < 0.001 ^‡^	*p* < 0.001 ^‡^	NA	NA	NA	NA	NA	NA
	Controls (n = 598)											
	Unstable Plaque (n = 166)	*p* < 0.001 ^‡^		*p* < 0.001 ^‡^	*p* < 0.001 ^‡^	*p* < 0.001 ^‡^	NA	NA	NA	NA	NA	NA
	Stable Plaque (n = 195)											
Del Porto, F. et al. [23]	A ^†^ (n = 20) vs. B ^†^ (n = 15)	t0 *:	*p* < 0.05 ^Δ¤^	NA	NA	*p* > 0.05 ^¤^	*p* > 0.05 ^¤^	*p* > 0.05 ^¤^	*p* > 0.05 ^¤^	NA	NA	NA
		t1 **:	*p* = 0.0047 ^Δ¤^	NA	NA	NA	NA	NA	NA	NA	NA	NA
	A ^†^ (n = 20) vs. C ^†^ (n = 16)	t0 *:	*p* < 0.001 ^Δ¤^	NA	NA	*p* < 0.001 ^¤^	*p* < 0.001 ^¤^	*p* < 0.05 ^¤^	*p* > 0.5 ^¤^	NA	NA	NA
	B ^†^ (n = 15) vs. C ^†^ (n = 16)	t0 *:	*p* > 0.05 ^¤^	NA	NA	*p* < 0.01 ^¤^	*p* < 0.01 ^¤^	*p* < 0.05 ^¤^	*p* < 0.01 ^¤^	NA	NA	NA
	A ^†^ (n = 20)	t0 *:	*p* = 0.0010 ^¤^	NA	NA	NA	NA	NA	NA	NA	NA	NA
		t1 **:										
	B ^†^ (n = 15)	t0 *:	*p* = 0.1563 ^¤^	NA	NA	NA	NA	NA	NA	NA	NA	NA
		t1 **:										
Zamani, M. et al. [24]	No IPN (n = 6)	*p* = 0.016 ^§^		NA	NA	*p* = 0.370, r = −0.173 ^œ^	NA	NA	NA	*p* = 0.765, r = 0.058 ^œ^	*p* = 0.577, r = −0.084 ^œ^	*p*= 0.242, r= −0.224 ^œ^
	Extensive IPN (n = 14)											
		*p* = 0.052 ^§^										
	Moderate IPN (n = 9)											
	Neovessel Count (n = 4.5 ^¥^)	*p* = 0.007, r = 0.491 ^œ^		NA	NA	*p* = 0.482, r =−0.173 ^œ^	NA	NA	NA	*p* = 0. 723, r = 0.069 ^œ^	*p* = 0.598, r = 0.106 ^œ^	*p* = 0.071, r = −0.340 ^œ^

* t0: just prior to carotid endarterectomy; ** t1: 30 min after carotid endarterectomy; ^Δ^ FGF-23 serum levels were only measured in 12 patients with complicated plaques and seven patients with uncomplicated plaques; ^†^ A: Complicated Plaques; B: Uncomplicated Plaques; C: Controls; ^‡^ FGF3 serum levels were measured using a second-generation C-terminal human enzyme-linked immunosorbent assay (Immutopics, San Clemente, CA, USA); ^¤^
*p* values from Kruskal–Wallis one way ANOVA followed by a Dunn posttest; ^§^
*p* values from post hoc pairwise comparison. ^œ^ coefficients of correlation calculated by the Spearman ρ correlation for scale variables; ^¥^ mean value of the number of neovessels observed within a 2-min Superb Microvascular Imaging (SMI) video clip. Legend—FGF-23: Fibroblast Growth Factor-23; HsCRP: High Sensitivity C Reactive Protein; OPG: Osteoprotegerin; IL: Interleukin; MCP-1: Monocyte Chemoattractant Protein-1; VEGF: Vascular Endothelial Growth Factor; TNF-a: Tumor Necrosis Factor-a; b-FGF: basic Fibroblast Growth Factor; ICAS: Internal Carotid Artery Stenosis; IPN: Intraplaque Neovascularization; NA: Non-Available Information; ^‡^ χ2 test for categorical values Δ FGF-23 serum levels were only measured in 12 patients with complicated plaques and 7 patients with uncomplicated plaques.

## Data Availability

No new data were created.

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
