# Peer review of "FGF-23 as a Biomarker for Carotid Plaque Vulnerability: A Systematic Review"

_medsci, 2025, doi:10.3390/medsci13010027_

Round 1
Reviewer 1 Report
Comments and Suggestions for Authors
A. Summary
The systematic review investigates the correlation of Fibroblast Growth Factor-23 (FGF-23) with plaque instability in coronary artery disease (CAD) patients during carotid endarterectomy (CEA). Three studies were evaluated to determine FGF-23 levels as a marker of unstable plaques and its value as a biomarker for the identification of plaque vulnerability. However, it needs minor and a few major changes or improvements for better understanding.
B. Strengths of the Manuscript:
- Relevance of the Topic:
- The research addresses the association of a developing biomarker—FGF-23—especially in CAD patients during CEA. This biomarker is significantly related to atherosclerosis pathogenesis in previous studies and is thus relevant in preventing stroke episodes if detected early.
- Methodology:
- The study followed PRISMA guidelines and registered in PROSPERO, bringing transparency to methodology.
- A wide search from databases such as MEDLINE, Scopus, Web of Science enhances dependability.
- NHLBI Study Quality Assessment Tool and GRADE criteria strengthens the data assessment.
- Data Presentation:
- Use of subgroup analyses brings more validity and applicability.
- Structured discussion with ample comparison of relevant studies.
- Critical Analysis & Limitations:
- The limitations were clearly elucidated such as small sample sizes, observational nature of the studies, and inability to adjust potential confounders.
C. Suggestions/ Areas of Improvement:
- Topic Title:
*(i) The title can be more precise and include the key findings to make it more impactful.
๐น Suggestions: (line 2 & 3)
Consider modifying the title as “FGF-23 as a Biomarker for Carotid Plaque Vulnerability: A Systematic Review.”
- Abstract:
*(i) Presence of heterogeneity among the studies is not discussed in the abstract
๐น Suggestions:(line 30)
Consider mentioning it in the abstract for estimating the effectiveness of the studies.
*(ii) Keywords selected may not reflect the study to 100%
๐น Suggestions: (line 40)
Consider keywords such as “Fibroblast Growth Factor-23, Carotid artery disease, Atherosclerosis, Carotid endarterectomy, Plaque instability, Inflammation, Neovascularization, Endothelial dysfunction, Chronic kidney disease, Superb Microvascular Imaging, Stroke risk, Restenosis.”
- Introduction:
*(i) Can be more focused by removing the CKD related FGF-23 information or provide a transition sentence connecting the FGF-23, CKD and atherosclerosis
๐น Suggestions: (line 73)
Can include the transition sentence such as “As Chronic kidney disease (CKD) progresses, phosphate excretion declines leading to compensatory rise in FGF-23 to maintain phosphate balance. As klotho (co-receptor of FGF-23) expression is decreased in CKD, persistent increase in FGF-23 levels contribute to endothelial damage and cardiovascular events.”
- Methods:
**(i) Exclusion criteria not described clearly in the methods sections.
๐น Suggestions:(line 100 & 101)
Include the exclusion criteria such as excluding animal studies, case studies, reviews, commentaries, editorials etc.
**(ii) Keywords, and Boolean operators used in search strategy are not mentioned in the main manuscript.
๐น Suggestions: (line 106 & 107)
Consider including them under the methods section although discussed in supplementary material to bring more transparency to the search strategy.
- Results:
*(i) All primary and secondary outcomes are described under the same “Main Findings” section.
๐น Suggestions: (line 191)
Consider discussing them separately as 3.3.1 Primary Outcomes and 3.3.2 Secondary Outcomes for a better comprehension of the key findings.
- Discussion & Conclusion:
*(i) Manuscript briefly mentions the lack of follow-up as a limitation, but it does not provide a detailed discussion on how this affects the study's conclusions.
๐น Suggestions: (line 440)
“A significant drawback of the studies included is the absence of long-term follow-up. Without monitoring FGF-23 levels after endarterectomy, it remains uncertain if this biomarker can predict restenosis, recurrent stroke, or long-term cardiovascular mortality. Upcoming research should include follow-up durations of no less than 6–12 months to evaluate the risk of restenosis and 1–5 years to examine the recurrence of strokes and survival results”
*(ii) Missing summary key points at the beginning of the discussion
๐น Suggestions: (line 353)
Consider including a brief summary of key findings at the beginning of discussion.
*(iii) Missing information on potential confounding factors, and if they were specifically adjusted or not and their implications if not adjusted.
๐น Suggestions:(line 431)
If not adjusted- “Potential confounding factors like diabetes mellitus, or chronic kidney disease, or hypertension, or vitamin D and calcium metabolism or medications varied among the study populations, potentially decreasing the ability to form independent conclusions in predicting the association of FGF-23 with plaque instability.”
*(iv) More clear mechanism of pathophysiology between the association of FGF-23 and plaque vulnerability is missing in discussion.
๐น Suggestions: (line 403 or wherever appropriate)
Consider starting as “FGF-23 can disrupt plaque stability through various mechanisms, leading to endothelial damage, infiltration of inflammatory cells, and neovascularization, elevating the risk of stroke and cardiovascular events…..” and then discuss each mechanism briefly.
**(ii) Explanation of heterogeneity of the studies is missing
๐น Suggestions: (line 421)
Consider explaining why there could be heterogeneity among the studies based on their respective data
*(v) Missing clarity in proposed future directions
๐น Suggestions:(line 445)
Consider “Study design involving Prospective, multicenter, 1–5 years follow-up studies, with patients stratified by CKD, diabetes, statin use, having primary endpoints, such as restenosis, stroke, and cardiovascular mortality, and secondary endpoints, such as endothelial dysfunction, inflammation, and intraplaque neovascularization, using multivariate adjustment and propensity score matching in statistical analysis should be encouraged which can help in gathering more information about clinical implications of FGF-23.”
- References
*(i) Some references missing italics and full names on journals.
๐น Suggestions:(line 483, 487, 492 & multiple locations)
Consider correcting Eur J Vasc Endovasc Surg to “European Journal of Vascular and Endovascular Surgery”
- Tables and Figures:
None.
D. Overall Recommendations:
Major: Marked as **
Minor: Marked as *
Author Response
Dear Professor Doctor Antoni Torres, Editor in Chief
Dear Gianluigi Li Bassi, Associate Editor
On behalf of my co-authors, I would like to thank you for the careful revisions and the opportunity to revise and improve our manuscript medsci-3501305 entitled “The Role of FGF23 in Carotid Endarterectomy: A Systematic Review of Mechanisms and Outcomes”
All authors agree with the changes made to reply to the reviewers' comments and are responsible for the data presented.
The manuscript is original, and no part has been published before, nor is any part under consideration for publication in another journal.
There are no conflicts of interest to declare.
After this revision, we believe that now the manuscript has the quality for publication in Medical Sciences.
Best regards,
João Rocha Neves, MD, MPH, PhD, FEBVS
Consultant of Angiology and Vascular Surgery
Invited Professor, Biomedicine Department – Unit of Anatomy, Faculty of Medicine, University of Porto, Portugal
Phone: (+351) 910486230
4200-319 Porto Portugal

Reviewer 2 Report
Comments and Suggestions for Authors
The manuscript (medsci-3501305) presents a systematic review examining the role of Fibroblast Growth Factor-23 (FGF-23) in carotid artery disease progression and its implications for carotid endarterectomy outcomes. The review provides a summary of the available literature and adheres to Preferred Reporting Items for Systematic Reviews and Meta-Analyses (PRISMA) guidelines, ensuring transparency in identification and selection of the studies. FGF-23 has been increasingly linked to atherosclerosis and increased cardiovascular risk. The manuscript effectively highlights the association between elevated serum FGF-23 levels and unstable carotid plaques, supporting the hypothesis that this potential biomarker may have clinical utility. I believe that this manuscript is suitable for publication, provided the following concerns are properly addressed.
Major concerns:
- In Figure 1, the PRISMA flow diagram shows that during the screening phase, reviewers examine 11 records and exclude 5 of them ("records excluded", n=5). Please specify the inclusion criteria applied during the screening phase and provide detailed reasons for the exclusion of the five records at this stage.
- After reading this manuscript, I am convinced that FGF-23 levels are correlated with plaque vulnerability in carotid artery disease. However, the mechanisms behind it have not been thoroughly studied, reviewed, or discussed. Therefore, the current title, 'The Role of FGF23 in Carotid Endarterectomy: A Systematic Review of Mechanisms and Outcomes' may not accurately reflect the content.
- Moreover, the causal relationship between elevated FGF-23 levels and plaque vulnerability in carotid artery disease has not been established. I recommend that the discussion section emphasizes the necessity for prospective, long-term studies to evaluate whether elevated FGF-23 levels can predict post-endarterectomy outcomes, thereby establishing FGF-23 as a reliable biomarker for carotid atherosclerosis.
Minor concerns:
- In the statistical results (e.g., Tables 3 and 4), reporting effect sizes—preferably standardized effect sizes—and their corresponding confidence intervals where possible would improve clarity and help the audience better understand and interpret the results across different studies.
- The results regarding inflammatory markers and plaque characteristics are inconsistent between studies 1-2 and study 3. The discussion would benefit from a deeper analysis of why these discrepancies exist and whether they are from methodological differences or true biological variation in the context of previous studies.
Author Response
Dear Professor Doctor Antoni Torres, Editor in Chief
Dear Gianluigi Li Bassi, Associate Editor
Medical Sciences
On behalf of my co-authors, I would like to thank you for the careful revisions and the opportunity to revise and improve our manuscript medsci-3501305 entitled “The Role of FGF23 in Carotid Endarterectomy: A Systematic Review of Mechanisms and Outcomes”.
All authors agree with the changes made to reply to the reviewers' comments and are responsible for the data presented.
The manuscript is original, and no part has been published before, nor is any part under consideration for publication in another journal.
There are no conflicts of interest to declare.
After this revision, we believe that now the manuscript has the quality for publication in Medical Sciences.
Best regards,
João Rocha Neves, MD, MPH, PhD, FEBVS
Consultant of Angiology and Vascular Surgery
Invited Professor, Biomedicine Department – Unit of Anatomy, Faculty of Medicine, University of Porto, Portugal
Phone: (+351) 910486230
4200-319 Porto Portugal

Round 2
Reviewer 1 Report
Comments and Suggestions for Authors
Greatly appreciate the authors' extra care in answering all the clarifications and modifying the responses accordingly. The manuscript is well-organized and comprehensive. I have no further recommendations.